# Discovering Irregularities from Computer Networks by Topological Mapping

Khalid Hamid [1], Muhammad Waseem Iqbal [2], Qaiser Abbas [3,4], Muhammad Arif [1], Adrian Brezulianu [5,6,*]
and Oana Geman [7]

1    Department of Computer Science, Superior University, Lahore 54000, Pakistan
2    Department of Software Engineering, Superior University, Lahore 54000, Pakistan
3    Faculty of Computer and Information Systems, Islamic University Madinah, Madinah 42351, Saudi Arabia
4    Department of Computer Science & IT, University of Sargodha, Sargodha 40100, Pakistan
5    Greensoft Ltd., 700050 Iaşi, Romania
6    Faculty of Electronics, Telecommunications and Information Technology, "Gheorghe Asachi" Technical
     University, 700050 Iaşi, Romania
7    Faculty of Electrical Engineering and Computer Science, The Computers, Electronics and Automation
     Department, Stefan cel Mare University of Suceava, 720229 Suceava, Romania
*    Correspondence: adi.brezulianu@greensoft.com.ro

**Abstract:** Any number that can be uniquely identified and varied by a graph is known as a graph invariant. This paper will talk about three unique variations of bridge networks, sierpinski networks, honeycomb, and hexagonal networks, with great capability of forecast in the field of software engineering, arithmetic, physics, drug store, informatics, and chemistry in setting with physical and chemical properties. Irregularity sombor invariant is newly introduced and has various expectation characteristics for various variations of bridge graphs or other networks, as mentioned. First, find the irregularities in the networks with the help of the Irregularity sombor index. This will be performed in a step by step procedure. The study will take an existing network, associate it with a graph after finding their vertices and edges, then solve the topology of a graph of a network. Graphical results demonstrate the upper and lower bounds and irregularities of certain networks, and mathematical results are used for modeling purposes. The review settled the topologies of graphs/networks of seven distinct sorts with an Irregularity sombor index. These concluded outcomes can be utilized for the demonstration and modeling of computer networks such as local area networks, Metropolitan area networks, Wide area networks, memory interconnection networks, processor interconnection networks, the spine of the internet, and different networks/designs of Personal computers, power generation networks, mobile base station and chemical compound amalgamation and so on.

**Keywords:** bridge networks; invariant; sierpinski; irregularity sombor index; maple; network graph; memory interconnection network

## 1. Introduction

Bridge graphs are introduced by T. Mansour and M. Schork, which is a combination of networks that bridge together in a single network [1]. A bridge graph is a graph obtained from the number of graphs G1, G2, G3 . . . .Gm by associating the vertices $v_i$ and $v_i + 1$ by an edge $\forall$, i = 1, 2 . . . m − 1 [2].

Sierpinski graphs comprise a broadly concentrated class of graphs of fractal nature appropriate in topology, mathematics of Tower of Hanoi, computer science, and somewhere else. An enormous number of properties such as physical-substance properties, thermodynamic properties, compound movement, and organic action, and so are not entirely set in stone by the synthetic utilization of graph theory. These properties can be described by specific graph invariants alluded to as topological indices. In QSAR/QSPR concentrating on these graphs, invariants play a crucial impact [3].

Honeycomb networks are used in IoT framework development. Wireless Sensor Network (WSN) is viewed as one of the crucial advances utilized in the Internet of things (IoT), consequently empowering different applications for doing continuous perceptions. Robot route in such networks was the primary inspiration for the presentation of the idea of tourist spots. A robot can recognize its area by conveying messages to obtain the distances between itself and the milestones. By believing networks to be a sort of graph, this idea was reclassified as a metric component of a graph which is the base number of hubs expected to recognize every one of the hubs of the graph. This thought reached out to the idea of the edge metric element of graph G, which is the base number of hubs required in a graph to distinguish each edge of the organization exceptionally. Ordinary plane networks can be handily built by rehashing customary polygons. This plan is of outrageous significance as it yields high generally speaking execution; thus, it very well may be utilized in different systems administration and IoT areas. The honeycomb and the hexagonal networks are two such famous lattice-inferred equal networks. In this paper, it is demonstrated that the base milestones expected for the honeycomb network HC($n$) and the hexagonal organization HX($n$) are 3 and 6 separately [4].

A Honeycomb network is a network in which $n$-number of layers of hexagonal networks exist and are utilized in versatile base station organization. Each cell has 6 vertices and 6 edges. Each cell is encircled by 6 other hexagonal cells, which makes it the design of the honeycomb network for the mobile base station and many chemical constructs.

The study presented the hexagonal network containing triangles, and such sorts of graphs are regularly known as oxide networks in numerical science. Different layers of triangles outline the simple development of the hexagonal network of aspect 2 HX(2), aspect 3 HX(3), and aspect 4 HX(4). Going on along these lines, by putting the furthest layer of triangles, we can obtain the hexagonal network HX($n$) with $n$ aspect [5].

Regular hexagonal networks are utilized in mobile base station networks, IoT networks, and numerous chemical compounds constructs with six sides continuously joined. Regular Hexagonal Cells Network has vertices of both degrees 2 and 3. By assessment, we obtain that it has 4 mn + 4m + m − 2 vertices and 6 mn + 5m + $n$ − 4 edges. There are three sorts of edges considering the degree of end vertices of each edge gives a definite portrayal of the edge set.

On the other hand, Gutman, in 2021, define the idea of sombor indices. A new vertex degree-based invariant graph named Sombor Index is used to capture the sharp lower and upper bounds of the connected network and the characteristics of the network reaching the boundaries [6]. V. R. Kulli derived a new irregularity index by taking an idea from sombor indices called the Irregularity sombor (ISO) index, which has the quality to predict irregularities present in computer networks [7].

For the most part, networks of various topologies are performed well and productive independently; however, the mix of at least two effectiveness split the difference. For the conventional explanation, the study talks about and settles the topology of bridge networks with the assistance of the graph hypothesis mathematically [8].

As one more emerging science is created with the assistance of computer sciences, mathematics, and chemistry called cheminformatics, whose critical sections integrate Quantitative structure-activity relationship (QSAR) and Quantitative structure-property relationships (QSPR), and the fragments can include the assessment of physicochemical attributes of manufactured combinations. QSAR is a modeling instrument used to settle the topology of networks or structure of mixtures and display the productive and best entertainer networks or structures. QSPR is likewise a modeling device that corresponds to the properties of a network structure with the assistance of numerical conditions or articulation. It additionally gives the quantitative relationship between the properties of networks or chemical structures. Points of topology as the numeric worth can be depicted with the assistance of a graph in light of invariance. It is finished because of the automorphism property of the graph. In the fields of computer sciences and chemistry, there are a ton of uses for graph hypothesis [9].

A topological index is arranged by changing a network structure into a number. Initially, our point is to present new computer models and networks that benefit from both effectiveness and advance with the help of topological indices. An interconnection network's structure can be numerically shown by a graph. The geography of a graph chooses the way where vertices are related by edges. From the geography of a network, specific properties can undoubtedly be settled. The greatest distance is settled between any two centers in the network. The degree of the center is recognized by the number of associations connected with it. Computer networks from intranet to overall networks, electric power interconnection, interpersonal organizations, the sexual affliction of networks of transmission, and genome networks are tantamount to graph hypothesis with the assistance of intricate networks examination contraption. This multitude of networks is at the top level of their utilization and enhanced. In this heap of cases, this study can register boundaries called Topological invariants (TIs) that numerically portray the connectedness plans (structure) between the center points or performers in a network. So, this study can develop a mind-stunning network of general arrangements of regulations accomplice regulations (centers) that direct regular natural subjects, for example. QSAR and QSPR are giving the establishment of these models. The last comment is that the use of the estimation in the network plane works with a quantitative assessment of different geography-protecting planning calculations [10].

This paper initially presents the issue articulation with a bridge graph and ISO index. In addition to auditing the writing, the third examines objectives, significance, research gap, and technique in the research system segment, the fourth area breaks down information, and the last area composes results and finishes up the research. The review has suggestions in the fields of computer science, physics, chemistry, mathematics, and bioinformatics for modeling reasons for networks, memory interconnections networks, power generation interconnection networks, and chemical mixtures. ISO invariant permits us to aggregate data about logarithmic structures and numerically foresee stowed-away properties of different structures, for example, certain computer networks.

Topological invariants draw us to collect information about logarithmic designs and give us a mathematical methodology to calculate the hidden properties of different certain computer networks and other constructs. Different techniques are accessible in history to look at the idea of a topological index. There are two chief contentions of topological indices; first one is the degree based topological, and the sub-optimal is known as distance-based topological indices. There are numerous such invariants are accessible in history. Irregularity Sombor Index has an extraordinary capacity of assumption in the field of computer science, math, chemistry, drugs, informatics, and power age in setting with physical and substance designs and organizations.

ISO index stands for Irregularity Sombor Index, which is introduced by V. R Kulli after taking inspiration from Sombor Indices. ISO index has the quality to predict the hidden properties of a network and find the lower bounds, upper bounds, and irregularities from the existing networks. The deduced results would be used for the modeling of certain computer networks, their gradation with best characteristics, finding new network architectures, and also used as guidelines for the developments of advanced networks used in different fields of computer science and other sciences.

## 2. Literature Review

The paper has shown that the star-like tree has the most ludicrous worth of the out-and-out irregularity index, and the caterpillar trees have the base worth of the full-scale irregularity index among all vertex trees with a proper number of regions. Anyway, it has been shown that the caterpillar trees of the most incredible degree three accomplish the base worth, and the trees with everything considered one developing vertex of degree more fundamental than three and containing essentially swinging and broadening vertices have the best [11].

This paper proposes evaluating the irregularity of a vector-respected morphological chief by the general opening between the summed-up proportion of pixel-wise distances and the Wasserstein metric. Other than presenting a degree of irregularity, suggested as the irregularity index, this paper additionally addresses its computational execution. Unequivocally, we see the best in general and the reasonable nearby irregularity indexes. The nearby irregularity index, which can be taken care of significantly more rapidly by amounting to expected gains of neighborhood windows, yields a lower bound for the general irregularity index. Computational primers with customary pictures outline the possibility of the proposed irregularity indexes [12].

Topological invariants enable us to gather information about logarithmic structures and give us a mathematical system to sort out the mysterious properties of different structures [13]. Different procedures are accessible in history to check the idea of a topological index [14]. There are two essential struggles of topological indices; first one is the degree based topological, and the sub-par is known as distance-based topological indices. There are many such invariants available in history [15,16]. ISO index has great capability of expectation in the field of computer science, mathematics, chemistry, pharmacy, informatics, and physics in setting with physical and chemical structures and networks [17].

Performing similar tests, a few prospects of developing novel degree-and distance-based graph irregularity indices are examined. By assessing the segregation capacity of various irregularity indices, it is illustrated (utilizing models) that in specific cases, two recently developed irregularity indices are more specific [18].

Dental impressions have been supposed to obtain fitting audit models. This framework is time-and turns out consuming for the orthodontist and could be weakening to the patient, especially when supports are fitted concerning an exploration project. This study wanted to assess the precision, steadfastness, and reproducibility of using intraoral photos and mortar models' photos in assessing Little's Irregularity Index (LII), tooth size-curve length blunder (TSALD), and Bolton's extents [19].

In this paper, the study describes the previously mentioned graphs with another option yet relatively straightforward methodology. Additionally, the review described the graphs having the greatest irregularity esteem among the classes Tn (Tricyclic graphs), TETn (Tetracyclic graphs), PNTn (Pentacyclic graphs), and HEXn (Hexacyclic graphs) [20].

The study describes irregularities of graphs and their conditions on the size boundaries as of late, standing out among mathematicians as well as hypothetical scientific experts. It is observed that these abnormalities are connected with the properties of the substance in question. Cerium oxide is an interesting earth metal formula, and it is a light yellow-white powder. In the current article, we are worried about processing the shut types of irregularity proportions of the general type of gem structure of Cerium Oxide in light of numerical model and computation [21].

The survey looks at a couple of related proportions of peripherality and centrality for vertices and edges in networks, including the Mostar index, which was introduced as a proportion of peripherality for the two edges and networks. The overview discredits a supposition on the best possible Mostar index of bipartite graphs. It asymptotically answers another issue on the biggest difference between the Mostar index and the inconsistency of trees. It, in a similar manner, shows different extremal limits and computational complexity results about the Mostar index, abnormality, and proportions of peripherality and centrality. The readings look at graphs where the Mostar index is certainly not an exact proportion of peripherality. It fosters a general gathering of graphs with the property that the Mostar index is imperative for edges that are closer to the center [22].

Graph indices have drawn extraordinary interest as they give us mathematical hints for a few properties of particles. Some indices give important data on the atoms viable utilizing numerical estimations, as it were. Consequently, the estimation and properties of graph indices have been the focal point of research. Normally, the qualities taken by a graph index are significant issues called the inverse issue. It requires information about the presence of a graph having an index equivalent to a given number. The inverse issue is

read up here for the Albertson irregularity index as a piece of examination on irregularity indices. A class of graphs is developed to Show that the Albertson index generally takes certain even integers. It has been demonstrated that there exists no less than one tree with an Albertson index equivalent to each even certain integer; however, 4. The presence of a unicyclic graph with an irregularity index equivalent to m is displayed for each even sure integer m aside from 4. It is likewise shown that the Albertson index of a cyclic graph can achieve any even certain integer [23,24].

We compare different studies with our present study in the following Table 1. After analysis, it found that as compared to others, we found irregularities in the existing networks, which was the major issue and hurdle in the efficiency, performance, and security of the networks. It also found sharp upper bounds and lower bounds of different networks.

**Table 1.** Analytical Comparison of Topological Invariants and their Applications.

| Sr. No. | Title of Research Paper | Year | Networks Solved | Invariants Used | Results |
|---------|------------------------|------|-----------------|-----------------|---------|
| 1 | Topological Properties Of Degree-Based Invariants Via M-Polynomial Approach | 2022 | Hexagonal Networks | Zagreb Indices, Randi'C, Product Connectivity Gourava Index and their Forms | Give valuable information about the molecular structure or network and applications in QSPR & QSAR. |
| 2 | Contraharmonic Quadratic Index Of Certain Nanostar Dendrimers | 2022 | Dendrimer Nanostars | Contraharmonic-Quadratic Index and Quadratic-Contraharmonic Index | computed the CQ index for some standard graphs |
| 3 | Some Results On The Sombor Indices of Graphs | 2021 | Degree-Regular Graph/Network | The Sombor Index, The Reduced Sombor Index and the Average Sombor Index | Establishing inequalities related to the aforementioned three graph invariants and proving a recently proposed conjecture concerning the sombor index |
| 4 | Some Basic Properties of Sombor Indices | 2021 | Regular Graph or Network | Vertex-Degree-Based (VDB) Molecular Structure Descriptors (Sombor Index and its Reduced Form) | Any reduced VDB index can be viewed as a reduced sombor-type index |
| 5 | Analysis Of Dendrimer Generation By Sombor Indices | 2021 | Dendrimers Generation Networks | Sombor Index and Reduced Sombor Index | Computed sombor indices for phosphorus-containing dendrimers & types of dendrimers. |
| 6 | Sombor Index of Some Nanostructures | 2021 | Nanostructures | Sombor Index | Computed explicit formulae for sombor index of 2D-lattice, nanotube, and nanotorus |
| 7 | Polynomials And General Degree-Based Topological Indices of Generalized Sierpinski Networks | 2021 | Sierpinski Networks | Connectivity Polynomials Such As $m$-Polynomial, Zagreb Polynomials, Forgotten Polynomial, ($A$, $B$)-Zagreb Index and Several Other General Indices | These facts can be Physicochemical properties of the molecules modeled on the $S(k, n)$ networks can be forecasted using the results. |

**Table 1.** *Cont.*

| Sr. No. | Title of Research Paper | Year | Networks Solved | Invariants Used | Results |
|---|---|---|---|---|---|
| 8 | The Calculations of Topological Indices on Certain Networks | 2021 | Hexagonal Networks | ABC Index, AZI Index, GA Index, The Multiplicative Version Of Ordinary First Zagreb Index, The Second Multiplicative Zagreb Index, and Zagreb Index | Calculating the correlation index provides potential help for scholars to study networks characteristics better. for further work, if the corresponding networks are replaced by other networks |
| 9 | Discovering Irregularities from Computer Networks by Topological Mapping | 2022 | Bridge Networs, Hexagonal Networks, Honeycomb Networks and Sierpinski Networks | Irregularity Sombor Index | Finding Sharp upper bounds, lower bounds and irregularities |

In geography, computer science, and related areas of mathematics, a topological property or Topological invariant (TI) is a property of a topological space that is invariant under homeomorphisms. On the other hand, a topological property is a legitimate class of topological spaces that are shut under homeomorphisms. That is, a property of spaces is a topological property if, at whatever point, a space X has that property. Each space homeomorphic to X has that property. Casually, a topological property is a property of the space that can be communicated utilizing open sets. These TI's are used to solve the network, and deduced results will be used for the modeling of the new network architectures.

Big data is working on the efficiency of accessing and transferring data fastly. The theme of this study is also the same to optimize the efficiency of the network through the evaluation of the topology of a network [25–29].

### 3. Research Methodology

#### 3.1. Objectives

The principal objective of this study is to research the anomalies in computer networks through topological invariants. The review figures out the force of earnestness of topological indices in specific computer networks such as computer networks, interconnection networks of processors, memory interconnection networks, mobile base station networks, power interconnection networks and chemical structures, and so on. In this paper, the study makes sense of the ISO index and its advantages. Its superb goal is to foster recipes, so it can look at the anomalies in the topology and execution of specific networks without doing/performing tests. The work derived a few outcomes which are utilized in the modeling of specific computer networks.

#### 3.2. Significance

The review is exceptionally critical these days since it makes mindfulness about irregular invariants of specific computer networks. It is likewise finding new and huge arrangements or expressions for the modeling of specific computer networks because no satisfactory arrangement has been tracked down till now because of its gradual and quick nature.

#### 3.3. Method

This methodical review will take a current bridge network or sierpiski network or hexagonal or honeycomb network, partner it using a graph, and tackle the physical layout of the graph with the assistance of the ISO index. The disturbing outcomes as recipes will contrast and existing outcomes. These derived outcomes will be pertinent to numerous

different networks in the fields of specific computer networks a short time later. This model is particularly used as it tackled the topologies of specific computer networks in numeric and graphical structure in step-by-step manner. At first, it takes existing networks, associates them with graphs, finds vertices and edges, converts them into a graph, solves the topology of a graph, deduces results in mathematical and graphical forms and at the end, applies them in different fields. These deduced results are also used in the modeling of existing networks for upgrading existing networks with the best characteristics and for creating new network architectures. It gives irregular results. After examination, a simulation instrument maple is utilized for the confirmation and approval of results. "Maple tool is a bunch of techniques to do limited geography. The methods were decided to outline essential topological developments and properties. Maple is a universally useful instrument for math, computer science, physics, information investigation, representation, and programming. It contains a huge number of specific capabilities that range from all areas of design. It is an emblematic and numeric processing climate as well as a multi-worldview programming language."

## 4. Experimental Results

A bridge graph is a network graph found from the number of network graphs $G_1$, $G_2$, $G_3$, ... $G_m$ by associating the vertices $v_i$ and $v_{i+1}$ by an edge $\forall$, i = 1,2, ... , m − 1.

$$ISO(G) = \sum_{ue} \sqrt{\left| d_u{}^2 - d_v{}^2 \right|} \tag{1}$$

Equation (1) shows the ISO index, which will be used for the solution of three variants of the bridge, honeycomb, hexagonal, and two variants of seirpinski networks. This ISO index equation is used for finding irregularities in the given networks mentioned before.

$$de = du + dv - 2$$

Table 2 describes the edge partitions of graph $G_r$ ($P_{s, v}$) over $P_s$ of the bridge graph given in Figure 1. It is shown four distinct types of edges with their frequencies and vertices in the above table. It explains Figure 1 of the bridge network.

**Table 2.** Edge partition of $G_r$ ($P_{s, v}$) over $P_s$.

| $\varepsilon$ | $\varepsilon(du, dv)$ | de | $\varepsilon(du, de)$ | Recurrence |
|---|---|---|---|---|
| $\varepsilon_1$ | $\varepsilon(1, 2)$ | 1 | $\varepsilon(1, 1)$ | R |
| $\varepsilon_2$ | $\varepsilon(2, 2)$ | 2 | $\varepsilon(2, 2)$ | 3r + 2 |
| $\varepsilon_3$ | $\varepsilon(2, 3)$ | 3 | $\varepsilon(2, 3)$ | R |
| $\varepsilon_4$ | $\varepsilon(3, 3)$ | 4 | $\varepsilon(3, 4)$ | r − 3 |

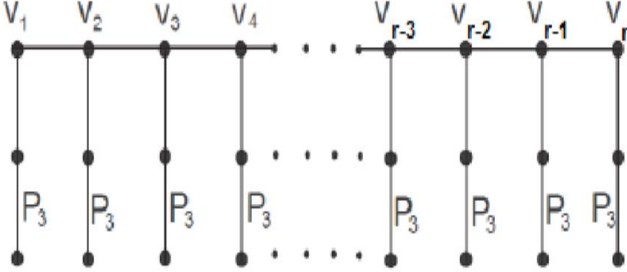

**Figure 1.** $G_r$ ($P_{s, v}$) over $P_s$ for Bridge Network.

*4.1. Main Results*

Figure 1 shows bridge networks in which bus networks and star networks bridge in a tree-like structure. The bridge network shows the distinct types of vertices and edges present in the above figure, which are also mentioned in Table 2.

### 4.1.1. Bridge Graph $G_r$ ($P_{s, v}$) over Path

If the vertex set is V, by the perception of Figure 2, it can arrange this vertex set into four subsetsV1, V2, V3, and V4, Such that V = V1 + V2 + V3 + V4. Assuming E addresses the edge set. Figure 2 shows that there are four unmistakable sorts of edges existing in the graph bridge graph $G_r$ ($P_{s, v}$) over the path of hybrid networks. Table 2 explains in detail the edges partition.

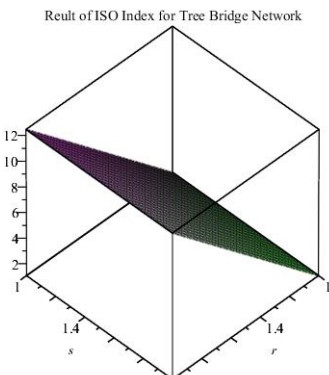

**Figure 2.** Result of ISO index for bridge networks of trees.

### 4.1.2. Theorem 1

Suppose *G* is a graph of $g_r$ ($p_{s, v}$) over $p_s$. Then, after investigation of bridge graphs by ISO index, the result is

$$\text{ISO} (G) = \sqrt{5}\, r \tag{2}$$

Equation (2) represents the proven results of the graph through edge partitions mentioned in Table 2 of $G_r$ ($P_{s, v}$) over $P_s$ mentioned in Figure 2.

Table 3 describes the edge partitions of graph $G_r$ ($K_{s, v}$) Over $K_s$ of the bridge graph given in Figure 3 with frequencies.

**Table 3.** Edge partition of $G_r$ ($K_{s, v}$) over $K_s$.

| $\varepsilon$ | $\varepsilon(du, dv)$ | de | $\varepsilon(du, de)$ | Recurrence |
|---|---|---|---|---|
| $\varepsilon_1$ | $\varepsilon_{(2, 2)}$ | 2 | $\varepsilon_{(2, 2)}$ | $rs - 2r$ |
| $\varepsilon_2$ | $\varepsilon_{(2, 3)}$ | 3 | $\varepsilon_{(2, 3)}$ | 4 |
| $\varepsilon_3$ | $\varepsilon_{(2, 4)}$ | 4 | $\varepsilon_{(2, 4)}$ | $2r - 4$ |
| $\varepsilon_4$ | $\varepsilon_{(3, 4)}$ | 5 | $\varepsilon_{(3, 5)}$ | 2 |
| $\varepsilon_5$ | $\varepsilon_{(4, 4)}$ | 6 | $\varepsilon_{(4, 6)}$ | $r - 3$ |

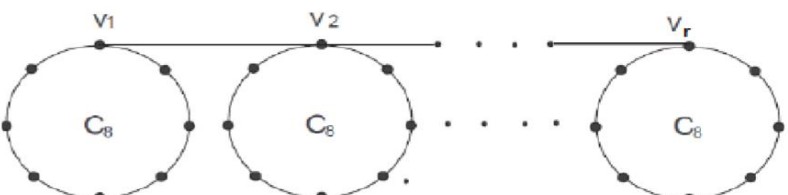

**Figure 3.** $G_r$ ($C_{s, v}$) over $C_s$ for Bridge Network.

### 4.2. Main Results

Figure 3 shows the bridge networks in which bus networks and ring networks bridge together.

### 4.2.1. Bridge Graph $G_r$ ($C_{s, v}$) over Cycle

Assuming V is the arrangement of vertices seen in Figure 3, this arrangement of vertices can be parted into four subclasses V = V1 + V2 + V3 + V4. When $\varepsilon$ addresses an

edge set. Figure 3 shows a half-and-half network cycle with five distinct kinds of edges in the network graph of the bridge graph $G_r$ ($C_s$, $_v$). Table 3 provides a detailed description of the edge set.

### 4.2.2. Theorem 2

Let $G$ be a graph of $G_r$ ($C_{s, v}$) over $C_s$, then after investigation of bridge graphs by ISO indices, the result is

$$\text{ISO}(G) = 8 + 4\sqrt{5} + 2\sqrt{3}(2r - 4) + 2\sqrt{5}(r - 3) \tag{3}$$

Equation (3) represents the proven results of the graph through edge partitions mentioned in Table 3 of $G_r$ ($C_{s, v}$) over the Cycle mentioned in Figure 3.

Figure 4 Shows the results of irregularities found in the topology of a cyclic bridge network through the ISO index.

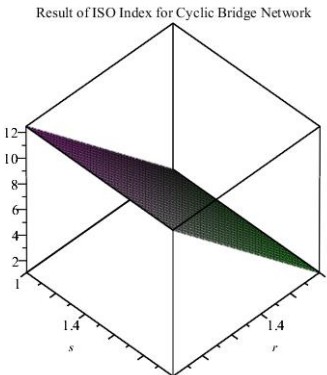

**Figure 4.** Result of ISO index for bridge network of multiple cycles.

Table 4 describes the edge partitions of graph $G_r$ ($C_{s, v}$) Over $C_s$ of the bridge graph given in Figure 4 with the number of occurrences.

**Table 4.** Edge partition of $G_r$ ($C_{s, v}$) over $C_s$.

| $\varepsilon$ | $\varepsilon(du, dv)$ | De | $\varepsilon(du, de)$ | Recurrence |
|---|---|---|---|---|
| $\varepsilon_1$ | $\varepsilon_{(4, 5)}$ | 7 | $\varepsilon_{(4, 7)}$ | 2 |
| $\varepsilon_2$ | $\varepsilon_{(4, S-1)}$ | $S + 1$ | $\varepsilon_{(4, s+1)}$ | 2 |
| $\varepsilon_3$ | $\varepsilon_{(5, 5)}$ | 8 | $\varepsilon_{(5, 8)}$ | $r - 2$ |
| $\varepsilon_4$ | $\varepsilon_{(5, S-1)}$ | $S + 2$ | $\varepsilon_{(5, s+2)}$ | $r - 2$ |
| $\varepsilon_5$ | $\varepsilon_{(S-1, S-1)}$ | $2s - 4$ | $\varepsilon_{(s-1, 2s-4)}$ | $[rs(r - 1) - 2(r + 1)]/2$ |

### 4.3. Main Results

Figure 5 shows the bridge networks in which bus networks and fully connected networks bridge together.

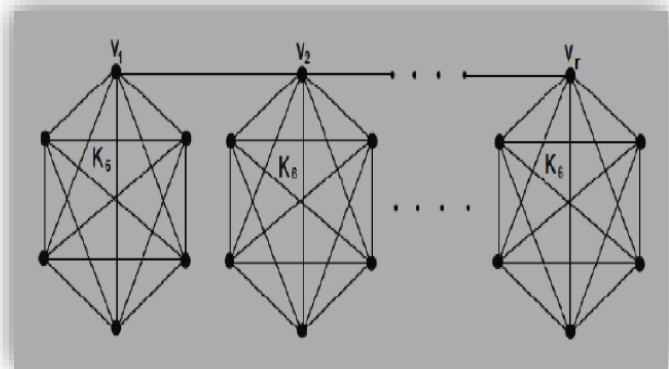

**Figure 5.** $G_r$ ($K_{s, v}$) over $K_s$.

### 4.3.1. Bridge Graph $G_r$ ($K_{s, v}$) over Complete Graph

Assuming that the vertices set are V, understanding Figure 5 allows us to sort this set of vertices into three subsets, V1, V2, and V3, so that V = V1 + V2 + V3. If E shows the edge set, Figure 5 displays the bridge network graph Gr (Ks, v) of the complete graph of the hybrid network. The bridge graph of the network graph has five different edges. Table 4 provides a detailed description of the edge set.

### 4.3.2. Theorem 3

Let *G* be a graph of $G_r$ ($K_{s, v}$) over $K_s$. then, after investigation of bridge graphs by ISO indices, the result is

$$\text{ISO (G)} = 6 + 2\sqrt{\left|-16 + (s-1)^2\right.} - \sqrt{\left|-25 + (s-1)^2\right.} \ (r-2) \tag{4}$$

Equation (4) represents the proven results of the graph through edge partitions mentioned in Table 4 of $G_r$ ($K_{s, v}$) over the complete graph mentioned in Figure 5.

Figure 6 Shows the results of irregularities found in the topology of a fully connected bridge network through the ISO index.

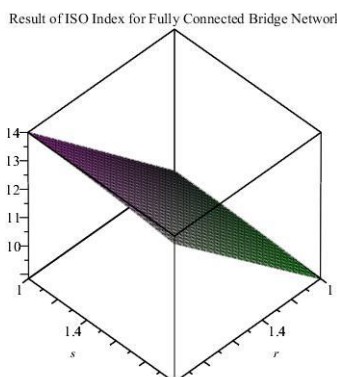

**Figure 6.** Result of ISO index for bridge network of multiple fully connected mesh.

Table 5 describes the edge partitions with their frequencies of the honeycomb network given in Figure 7. The study found vertices of G are either 5, 7, or 9 shown in Table 5 and Figure 7 also. After the calculation, the number of edges formed is 6, $12(n-1)$, $6(n-1)$, and $9n2 - 21n + 12$, shown in the above table.

**Table 5.** Edge partition of honeycomb network.

| E | ε(du, dv) | de | ε(du, de) | Recurrence |
|---|---|---|---|---|
| $\varepsilon_1$ | $\varepsilon(5, 5)$ | 8 | $\varepsilon(5, 8)$ | 6 |
| $\varepsilon_2$ | $\varepsilon(5, 7)$ | 10 | $\varepsilon(5,10)$ | $12(n - 1)$ |
| $\varepsilon_3$ | $\varepsilon(7, 9)$ | 14 | $\varepsilon(7, 14)$ | $6(n - 1)$ |
| $\varepsilon_4$ | $\varepsilon(9, 9)$ | 16 | $\varepsilon(9, 16)$ | $9n^2 - 21n + 12$ |

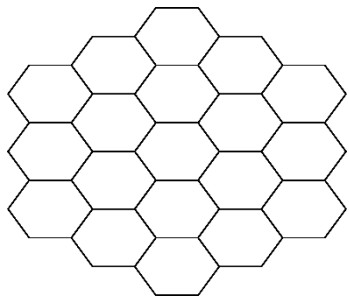

**Figure 7.** Honeycomb network with *n* number of cells ($HC_n$).

*4.4. Main Results of Honeycomb Network*

Figure 7 shows honeycomb networks in which *n*-number of layers of hexagonal networks exist and are used in mobile base station networks. Every cell has six vertices and six edges. Every cell is surrounded by six other hexagonal cells, which makes it the structure of the honeycomb network for the mobile base station.

### 4.4.1. Honeycomb Graph

Let G = $HC_n$ the *n*-layered honeycomb network with *n* hexagons among middle and boundary hexagons by $HC_n$. $HC_n$ is developed by adding a layer of hexagon around $HC_{n-1}$ (see Figure 7). The order and size of $HC_n$ are $6n^2$ and $9n^2 - 3n$, separately. Table 5 explains in detail the edges partition.

### 4.4.2. Theorem 4

Let *G* be a graph of a honeycomb, then, after investigation of honeycomb graphs by ISO index

$$\text{ISO (G)} = 2\sqrt{6}(12n - 12) + 4\sqrt{2}\,(6n - 6) \tag{5}$$

Equation (5) represents the proven results of the graph through edge partitions mentioned in Table 5 of the honeycomb mentioned in Figure 4.

Figure 8 Shows the results of irregularities found in the topology of a honeycomb network through the ISO index.

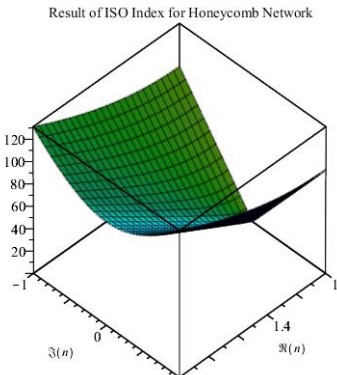

**Figure 8.** Result of ISO index for honeycomb network.

Table 6 describes the edge partitions with frequencies regarding them in the graph of the regular hexagonal cells network given in Figure 9 with frequencies. The study found vertices of *G* are either 2 or 3 shown in Table 6 and Figure 9 also. After the calculation, the number of edges formed is 2n + 4, 4m + 4n + 4, and 6 mn + m − 5n − 4, shown in the above table.

**Table 6.** Edge partition of regular hexagonal cells network.

| E | ε(du, dv) | de | ε(du, de) | Recurrence |
|---|---|---|---|---|
| $\varepsilon_1$ | $\varepsilon_{(2, 2)}$ | 2 | $\varepsilon_{(2, 2)}$ | 2n + 4 |
| $\varepsilon_2$ | $\varepsilon_{(2, 3)}$ | 3 | $\varepsilon_{(2, 3)}$ | 4m + 4n + 4 |
| $\varepsilon_3$ | $\varepsilon_{(3,3)}$ | 4 | $\varepsilon_{(3, 4)}$ | 6 mn + m − 5n − 4 |

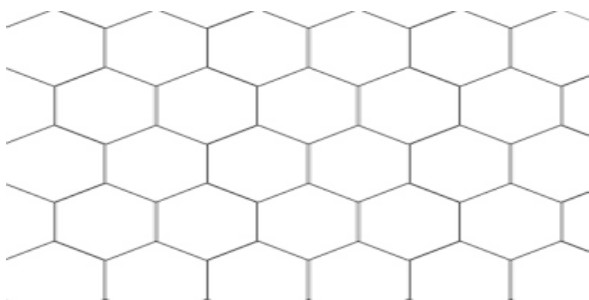

**Figure 9.** Regular hexagonal cells network.

*4.5. Main Results of Regular Hexagonal Cell Network*

Figure 9 shows regular hexagonal networks used in mobile base station networks with six sides consecutively attached.

### 4.5.1. Regular Hexagonal Cells Network

Let $G = RH_{(m, n)}$. The vertices of *G* are both of degree 2 or 3, as referenced in Figure 9. By estimation, we obtain that *G* has 4 mn + 4m + m − 2 vertices and 6 mn + 5m + *n* − 4 edges. In *G*, there are three kinds of edges in light of the level of end vertices of each edge Table 6 provides a detailed description of the edge set.

### 4.5.2. Theorem 5

Let *G* be a graph of the regular hexagonal network, then, after investigation of honeycomb graphs by ISO index.

$$\text{ISO (G)} = \sqrt{5}(4m + 4n − 4) \tag{6}$$

Equation (6) represents the proven results of the graph through edge partitions mentioned in Table 6 of regular hexagonal mentioned in Figure 9. with the help of the ISO Index.

Figure 10 Shows the results of irregularities found in the topology of a hexagonal network through the ISO index.

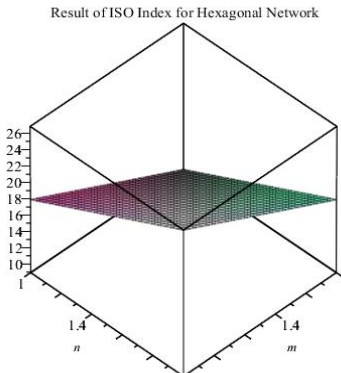

**Figure 10.** Result of ISO index for the hexagonal network.

Table 7 describes the edge partitions of graph G of the Sierpinski graph with frequencies given in Figure 11.

**Table 7.** Edge partition of Sierpinski ($S_n$).

| E | $\varepsilon(du, dv)$ | De | $\varepsilon(du, de)$ | Recurrence |
|---|---|---|---|---|
| $\varepsilon_1$ | $\varepsilon(2, 4)$ | 4 | $\varepsilon(2, 4)$ | 6 |
| $\varepsilon_2$ | $\varepsilon(4, 4)$ | 6 | $\varepsilon(4, 6)$ | $3^n - 6$ |

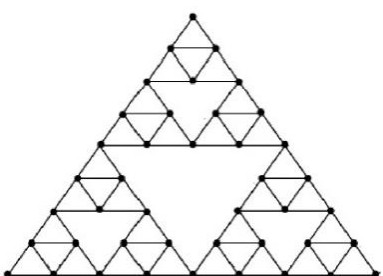

**Figure 11.** Sierpinski Network ($S_n$).

### 4.6. Main Results

Figure 11 shows Sierpinski Network with dimension ($S_n$), which is used in computer science and electronics for applying the loop concept. It is very much effective in integrated circuits, memory interconnection networks, power generation interconnection networks, etc.

#### 4.6.1. Sierpinski Network Graph

Suppose E (G) symbolizes the set of edges. Figure 11 shows two distinct kinds of edges existing in the network graph of Sierpinski. Table 7 explains in detail the edges partition.

#### 4.6.2. Theorem 6

Let G be a graph of $S_n$, then, after investigation of $S_n$ graphs by ISO index

$$\text{ISO (G)} = 12\sqrt{3} \tag{7}$$

Equation (7) represents the proven results of the graph through edge partitions mentioned in Table 7 of $S_n$ mentioned in Figure 11. with the help of the ISO index for the sake of improvements in existing networks and the development of new architectures.

Figure 12 shows the results of irregularities found in the topology of a Sierpinski network through the ISO index.

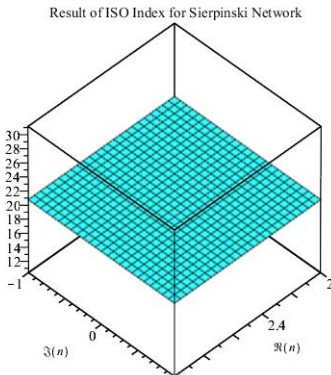

**Figure 12.** Result of ISO index for Sierpinski network $S_n$.

Table 8 describes the edge partitions of graph G of the Sierpinski network graph given in Figure 13 with frequencies.

**Table 8.** Edge partition of Sierpinski Network $S_{(n, k)}$.

| $\varepsilon$ | $\varepsilon(du, dv)$ | De | $\varepsilon(du, de)$ | Recurrence |
|---|---|---|---|---|
| $\varepsilon_1$ | $\varepsilon_{(2, k)}$ | k | $\varepsilon_{(2, k)}$ | 2k |
| $\varepsilon_2$ | $\varepsilon_{(3, 3)}$ | 4 | $\varepsilon_{(3, 4)}$ | $(k^{n+1} - 5k)/2$ |

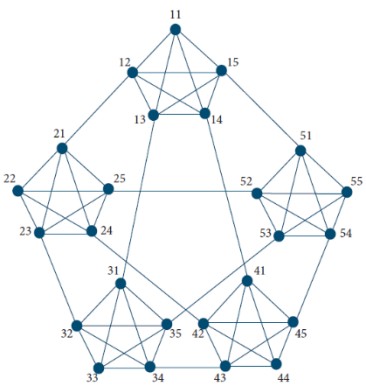

**Figure 13.** Sierpinski Network $S_{(n, k)}$.

*4.7. Main Results*

4.7.1. Sierpinski Network $S_{(n, k)}$

Suppose E (G) characterizes the set of edges Figure 13 shows that there are two distinct classes of edges existing in the Sierpinski network graph of $S_{(n, k)}$. Table 8 explains in detail the edges partition.

Figure 13 shows the Sierpinski Network $S_{(n, k)}$ with two distinct edges named '*n*' and 'k' for generalizing the network or graph.

4.7.2. Theorem 7

Let *G* be a graph of $S_{(n, k)}$, then after investigation of $S_{(n, k)}$ graphs by *ISO* index

$$ISO\,(G) = 2\sqrt{\left|k^2 - 4\right|}\,k \tag{8}$$

Equation (8) represents the proven results of the graph through edge partitions mentioned in Table 8 of $S_{(n, k)}$ mentioned in Figure 13 with the help of the *ISO* index.

Figure 14 shows the results of irregularities found in the topology of a Sierpinski $S_{(n, k)}$ network through the *ISO* index.

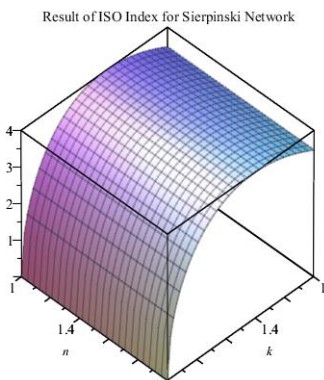

**Figure 14.** Result of ISO index for Sierpinski network $S_{(n, k)}$.

## 5. Conclusions

TIs have loads of purposes and executions in many fields of computer science, chemistry, physics, informatics, mathematics, material sciences, and more. Be that as it may, the greatest possible level of huge application is in the non-definite QSPR and QSAR. TIs are related to the structure of networks, the spine of the web, LANs, memory structure, mobile base station structure, and chemical structure. However, the present article talks about the ISO index, which is newly introduced and has various expectation qualities in setting anomalies in the topologies of the specific computer networks. The review found anomalies in the topologies of the various variations of bridge graphs or networks, i.e., $G_r$ (Ps, v), $G_r$ (Cs, v) and $G_r$ (Ks, v), honeycomb network $HC_n$, hexagonal network $H_{(m, n)}$, Sierpinski networks, i.e., $S_n$, $S_{(n, k)}$ networks. Figure 2, Figure 4, Figure 6, Figure 8, Figure 10, Figure 12, and Figure 14 give the graphical representation of the ISO index for the above-mentioned graphs of networks. Irregularity Sombor Index found lower bounds, upper bounds, and irregularities of all mentioned networks well prediction quality of best characteristic. These reasoned outcomes will be utilized for the modeling of computer networks (like LAN, MAN, WAN, and the spine of the web), mobile, base station power generation interconnection networks, memory interconnection networks, processor interconnection networks, chemical structures, picture handling, bioinformatics, and so on.

**Author Contributions:** Conceptualization, K.H.; Methodology, K.H. and M.W.I.; Formal analysis, K.H. and A.B.; Investigation, O.G.; Resources, M.A. and O.G.; Data curation, M.W.I. and A.B.; Writing—original draft, K.H.; Writing—review & editing, M.W.I.; Supervision, M.W.I. and M.A.; Project administration, O.G.; Funding acquisition, Q.A. All authors have read and agreed to the published version of the manuscript.

**Funding:** This work was funding supported by the Programme: SME Growth Romania–Priority ICT, Contract Greensoft No. 2020/548467, Project name: AI FERODATA.

**Acknowledgments:** We are thankful to our relations and associates who gave us moral help.

**Conflicts of Interest:** The creators pronounce that they have no irreconcilable situations to report in regard to the current review.

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
