# Peer review of "Discovering Irregularities from Computer Networks by Topological Mapping"

_applsci, doi:10.3390/app122312051_

Round 1

Reviewer 1 Report

1. Add a brief finding of results at the end of the abstract

2. Add significance clearly at the end of the introduction section

3. Do Authors need to explain ISO?

4. Describe Hexagonal & honeycomb networks?

5. In Methodology: Why use this model?

6. Figure numbers and table numbers are not correct

7. All equations need more explanation

8. Describe the tool used for optimization and validation.

9. How important are Topological invariants (TIs)? Please explain within the article

10. Describe Hexagonal and honeycomb within the article.

11. Why use this model in the methodology section?

12. Table 1 and figure 1 need more explanation.

13. Clearly explain Equations 10 and 11.

Author Response

Reviewer 1

Dear Reviewer,

Thank you very much for giving us the opportunity to revise the manuscript. We would like to thank the editor and all the reviewers for their valuable comments and suggestions. Based on the feedback, we have revised our manuscript. We marked the revisions in the manuscript as red color. For clarity, we have marked our responses in blue. We also highlight the revised content in the draft to facilitate the reviewer and editor.

Comment 1: Add a brief finding of results at the end of the abstract

Response 1: We have been added finding results in abstract section in this manner “At first find the irregularities in the networks with the help of ISO index. Graphical results demonstrate the upper, lower bounds and irregularities of the certain networks and mathematical results used for modeling purpose.

Comment 2: Add significance clearly at the end of the introduction section.

Response 2: We have write-up its main theme in introduction section with the following words as “Topological invariants draw in us to collect information about logarithmic designs and give us a mathematical methodology to calculate the hidden properties of different certain computer networks and other constructs. Different techniques are accessible history to really look at the idea of a topological index. There are two chief contentions of topological indices, first one is the degree based topological and the sub optimal is known as distance based topological indices. There are numerous such invariants are accessible in history [24]. Irregularity Sombor Index has extraordinary capacity of assumption in the field of computer science, math, chemistry, drugs, informatics and power age in setting with physical and substance designs and organizations Significance is also mentioned in Chapter 3 section 3.2 Research Methodology “The review is exceptionally critical these days since it makes mindfulness about irregularity invariants of specific computer networks. It is likewise finding new and huge arrangements or expressions for the modeling of specific computer networks because no satisfactory arrangement has been tracked down till now because of its gradual and quick nature.

Comment 3: Do Authors need to explain ISO?.

Response 3: It is defined in the introduction section as “ISO index stands for Irregularity Sombor Index which is introduced V. R Kulli after taking inspiration form Sombor Indices. ISO index has quality to predict the hidden properties of a network and find the lower bounds, upper bounds and irregularities from the existing networks.

Comment 4: Describe Hexagonal & honeycomb networks?

Response 4: We described in introduction section as “Honeycomb networks is a network in which n-number of layers of hexagonal networks are exist and utilized in versatile base stations organization. Each cell has 6 vertices and 6 edges. Each cell is encircled by 6 other hexagonal cells which make it the design of the honeycomb network for the mobile base station and many chemical constructs.” and “Regular hexagonal networks utilized in mobile base stations network, IoT networks and numerous chemical compound constructs with six sides continuously joined. Regular Hexagonal Cells Network have the vertices of both degree 2 or 3, as referred to in Figure 9. By assessment, we get that it has 4mn + 4m + m - 2 vertices and 6mn + 5m + n - 4 edges. There are three sorts of edges considering the degree of end vertices of each edge Tab. 5 gives a definite portrayal of the edge set.”

Comment 5: In Methodology: Why use this model?.

Response 5: We use this model particularly because of following reason “This model is particularly used as it tackled the topologies of specific computer networks in numeric and graphical structure in step by step manner. It gives exact irregularity results. After examination, a simulation instrument maple is utilized for the confirmation and approval of resultsAt first it takes existing networks, associate it with graph, finds vertices and edges, convert it into graph, solve the topology of a graph, deduced results in mathematical and graphical forms and at the end apply them in different fields. These deduced results are also used in the modeling of existing networks for upgrading existing networks with best characeristics and for creating new network architectures. We mentioned in the Method section.

Comment 6: Figure numbers and table numbers are not correct

Response 6: Thank you figure number 2 is wrongly mentioned now correct it.

Comment 7: All equations need more explanation

Response 7: It has been done in this manners Eq. (1) shows the ISO index which will be used for the solution of three variants of bridge, honeycomb, hexagonal and two variants of seirpinski networks mentioned in Fig. 1, 3, 5, 7,9 ,11 and 13 respectively. This ISO index equation is used for finding irregularities of the given networks mentioned before. Eq. (1) is the equation all others are results of this equation in context with different certain networks.

Comment 8: Describe the tool used for optimization and validation.

Response 8: It has been done in this manners “Maple tool is a bunch of techniques to do limited geography. The methods were decided to outline essential topological developments and properties. Maple is a universally useful instrument for math, computer science, physics, information investigation, representation, and programming. It contains huge number of specific capabilities that range all areas of designing. It is an emblematic and numeric processing climate as well as a multi-worldview programming language”

Comment 9: How important are Topological invariants (TIs)? Please explain within the article

Response 9: It is explained at the end of section Literature Review as “In geography, computer science and related areas of mathematics, a topological property or Topological invariant (TI) is a property of a topological space that is invariant under homeomorphisms. On the other hand, a topological property is a legitimate class of topological spaces which is shut under homeomorphisms. That is, a property of spaces is a topological property if at whatever point a space X has that property each space homeomorphic to X has that property. Casually, a topological property is a property of the space that can be communicated utilizing open sets. These TI's are used to solve the network and deduced results will be used for the modeling of the new network architectures.”

Comment 10: Describe Hexagonal and honeycomb within the article.

Response 10: We described in introduction section as “Honeycomb networks is a network in which n-number of layers of hexagonal networks are exist and utilized in versatile base stations organization. Each cell has 6 vertices and 6 edges. Each cell is encircled by 6 other hexagonal cells which make it the design of the honeycomb network for the mobile base station and many chemical constructs.” and “Regular hexagonal networks utilized in mobile base stations network, IoT networks and numerous chemical compound constructs with six sides continuously joined. Regular Hexagonal Cells Network have the vertices of both degree 2 or 3, as referred to in Figure 9. By assessment, we get that it has 4mn + 4m + m - 2 vertices and 6mn + 5m + n - 4 edges. There are three sorts of edges considering the degree of end vertices of each edge Tab. 5 gives a definite portrayal of the edge set.”

Comment 11: Why use this model in the methodology section?

Response 11: We use this model particularly because of following reason “This model is particularly used as it tackled the topologies of specific computer networks in numeric and graphical structure in step by step manner. It gives exact irregularity results. After examination, a simulation instrument maple is utilized for the confirmation and approval of resultsAt first it takes existing networks, associate it with graph, finds vertices and edges, convert it into graph, solve the topology of a graph, deduced results in mathematical and graphical forms and at the end apply them in different fields. These deduced results are also used in the modeling of existing networks for upgrading existing networks with best characeristics and for creating new network architectures.

Comment 12: Table 1 and figure 1 need more explanation.

Response 12: It is explained in the section 4.1.1  but we also explain in the following manners “Tab. 1 describes the edge partitions of graph Gr (Ps, v) over Ps of the bridge graph given in Fig. 1. It is shown four distint types of edges with their frequenies and vertices in the above table. It is basically explaining the figure 1 of bridge network.” And “Fig. 1 shows bridge networks in which bus networks and star networks bridge in a tree-like structure. The bridge network showing the distinct types of vertices and edges present in the above figure which are also mentioned in the Tab. 1.

Comment 13: Clearly explain Equations 10 and 11.

Response 13: Thank you so much but Equations 10 and 11 are not there in the manuscript.

Reviewer 2 Report

Very interesting ant timely article. I think it deserves publication and I am recommending accept with minor corrections. But there are some minor issues that require your attention. I list these corrections below as feedback / comments, and I am looking forward to reading the updated version of this article. 

- you should take out abbreviations from the abstract. The abstract is for general audiences, and you should start with abbreviations in the introduction chapter. 

- there are occasional errors in the text, or at least they appear as errors, maybe they are part of the formulas, but its worth checking the text for small errors in the text, e.g., edge∀, irrt esteem, etc. 

- if these are not errors in the text, then maybe you should include a formula key that would include all terms, before your start with using them. Or maybe there are just small errors. You need to check. 

- how would these computer irregularities be affected by new forms of data? and/or new AI algorithms? There are recent articles on this topic that review recent and relevant literature, for example, on the related topic of ‘New and emerging forms of data and technologies’ - see: https://doi.org/10.1007/s11042-022-13451-5 and on the related topic of ‘the ‘future values and risks from artificial intelligence’ - see: https://doi.org/10.1007/s12553-022-00691-6 - It would be interesting to see a few sentences reviewing and comparing your work in relations to these recent studies in related topics.

One final comment, you should check if all the things discussed in the introduction, are also discussed in the conclusion. because the introduction is much longer than the conclusion. 

- Good luck with the corrections and I am looking forward to reading the updated version of your article. 

Author Response

Dear Reviewer,

Thank you very much for giving us the opportunity to revise the manuscript. We would like to thank the editor and all the reviewers for their valuable comments and suggestions. Based on the feedback, we have revised our manuscript. We marked the revisions in the manuscript as red color. For clarity, we have marked our responses in blue. We also highlight the revised content in the draft to facilitate the reviewer and editor.

Comment 1: Very interesting ant timely article. I think it deserves publication and I am recommending accept with minor corrections. But there are some minor issues that require your attention. I list these corrections below as feedback / comments, and I am looking forward to reading the updated version of this article. 

Response 1: Thank you very it is highly encouraging for us

Comment 2: you should take out abbreviations from the abstract. The abstract is for general audiences, and you should start with abbreviations in the introduction chapter. 

Response 2: We have removed all abbreviations according to your instructions. Now the abstract loos like this “Any number that can be uniquely identified and varied by a graph is known as a graph invariant. This paper will talk about three unique variations of bridge networks, sierpinski networks, honeycomb and hexagonal networks with great capability of forecast in the field of software engineering, arithmetic, physics, drug store, informatics and chemistry in setting with physical and chemical properties. Irregularity sombor invariant is newly introduced and has various expectation characteristics for various variations of bridge graphs or other networks as mentioned. First, find the irregularities in the networks with the help of the Irregularity sombor index. Graphical results demonstrate the upper, lower bounds and irregularities of certain networks and mathematical results are used for modeling purposes. The review settled the topologies of graphs/networks of seven distinct sorts with Irregularity sombor index. These concluded outcomes can be utilized for the demonstration of computer networks like Local area network, Metropolitan area network, and Wide area network, memory interconnection networks, processor interconnection networks, the spine of the internet and different networks/designs of Personal computers, power generation networks, mobile base station and chemical compound amalgamation and so on

Comment 3: there are occasional errors in the text, or at least they appear as errors, maybe they are part of the formulas, but its worth checking the text for small errors in the text, e.g., edge∀, irrt esteem, etc.

Response 3: Infact “irrt” used for IRREGULARITY, but now it is written with full spellings.  This one ‘∀’ is a symbol used for universal quantifier but it required a space between edge and symbol like this “edge ∀”. It has been corrected as “Bridge graphs are introduced by T. Mansour and M. Schork which is a combination of networks that bridge together in a single network [1]. A bridge graph is a graph obtained from the number of graphs G1, G2, G3,...Gm by associating the vertices vi and vi + 1 by an edge ∀, i = 1,2,..., m − 1 [2].

Comment 4: if these are not errors in the text, then maybe you should include a formula key that would include all terms, before your start with using them. Or maybe there are just small errors. You need to check. 

Response 4: These all are checked and removed all errors if exist like this “In this paper, the study describes the previously mentioned graphs with another option yet relatively straightforward methodology. Additionally, the review described the graphs having the greatest irregularity esteem among the classes Tn (Tricyclic graphs), TETn (Tetracyclic graphs), PNTn (Pentacyclic graphs) and HEXn (Hexacyclic graphs) [20].

Comment 5: how would these computer irregularities be affected by new forms of data? and/or new AI algorithms? There are recent articles on this topic that review recent and relevant literature, for example, on the related topic of ‘New and emerging forms of data and technologies’ - see: https://doi.org/10.1007/s11042-022-13451-5 and on the related topic of ‘the ‘future values and risks from artificial intelligence’ - see: https://doi.org/10.1007/s12553-022-00691-6 - It would be interesting to see a few sentences reviewing and comparing your work in relations to these recent studies in related topics.

Response 5: We have studied and conclude that these above studies and this manuscript study both are working for the same cause mentioned but our study is working on topological context at the end of Literature Review section as “Big data is working on the efficiency of accessing and transferring of data fastly, the theme of this study is also same to optimized the efficiency of network but through evaluation of topology of a network [25][26][27].

Comment 6: One final comment, you should check if all the things discussed in the introduction, are also discussed in the conclusion. because the introduction is much longer than the conclusion. 

Response 6: Yes we have checked all the introduction and conclusion and found all the things both in introduction and conclusion except one which we are mentioning here and conclusion also as “Irregularity Sombor Index found lower bounds, upper bounds and irregularities of all mentioned networks well prediction quality of best characteristics”.

Comment 7: Good luck with the corrections and I am looking forward to reading the updated version of your article. 

Response 7: Thank you very it is highly encouraging and fruitful for us

Reviewer 3 Report

In this paper, the authors investigated about three unique variations of bridge networks, sierpinski networks, honeycomb and hexagonal networks with great capability of forecast in the field of software engineering, arithmetic, physics, drug store, informatics and chemistry in setting with physical and chemical properties. ISO invariant is newly introduced and has various expectation characteristics for various variations of bridge graphs or other networks as mentioned.The manuscript is interesting, which can be considered to published if some parts have been revised, the comments are follows

(1) Please polish the abstract. Please check the logic of abstract. Please add sentences to explain the meaning, the main points, the improvement and the promising application of the study. Plenty of detail data have given, however, in abstract, important procedures and results should be mentioned in simple manner. Please focus on the main points and the improvement of the study.

(2) Please highlight the advance of the study in Introduction. Please explain the development and creative work. The literature review should be carefully considered.

(3)The authors also should do a comparison with the previous reports to the advantage of this manuscript.

(4) Fig.5 is not clear, and the authors can consider to replote it.

(5) The authors can do a short introduction about the development in data acquisition or sensing. Especially, the topology technology is used in the sensing. Some references can be considered to add in this manuscript. For instance,

#B.F. Wan et al. in IEEE Sensors Journal, vol. 21, no. 1, pp. 331-338, 1 Jan.1, 2021, doi: 10.1109/JSEN.2020.3013289.

Author Response

Manuscript ID: applsci-1994033

Title: Discovering Irregularities from Computer Networks by Topological Mapping

Reviewer 3

Dear Reviewer,

Thank you very much for giving us the opportunity to revise the manuscript. We would like to thank the editor and all the reviewers for their valuable comments and suggestions. Based on the feedback, we have revised our manuscript. We marked the revisions in the manuscript as red color. For clarity, we have marked our responses in blue. We also highlight the revised content in the draft to facilitate the reviewer and editor.

Comment 1: In this paper, the authors investigated about three unique variations of bridge networks, sierpinski networks, honeycomb and hexagonal networks with great capability of forecast in the field of software engineering, arithmetic, physics, drug store, informatics and chemistry in setting with physical and chemical properties. ISO invariant is newly introduced and has various expectation characteristics for various variations of bridge graphs or other networks as mentioned. The manuscript is interesting, which can be considered to published if some parts have been revised, the comments are follows. 

Response 1: Thank you very much, this appreciation is highly encouraging for us

Comment 2: (1) Please polish the abstract. Please check the logic of abstract. Please add sentences to explain the meaning, the main points, the improvement and the promising application of the study. Plenty of detail data have given, however, in abstract, important procedures and results should be mentioned in simple manner. Please focus on the main points and the improvement of the study. 

Response 2: We have added the working procedure and implications in the abstract section to strengthen the logic of the abstract as “First, find the irregularities in the networks with the help of the Irregularity sombor index. This will be done step by step procedure. The study will take an existing network, associate it with graph after finding their vertices and edges, then solve the topology of a graph of network. Graphical results demonstrate the upper, lower bounds and irregularities of certain networks and mathematical results are used for modeling purposes. The review settled the topologies of graphs/networks of seven distinct sorts with Irregularity sombor index. These concluded outcomes can be utilized for the demonstration and modeling of computer networks” 

Comment 3: (2) Please highlight the advance of the study in Introduction. Please explain the development and creative work. The literature review should be carefully considered.

Response 3: Advancement and more valuable work highlights are explaining like that “Topological invariants draw us to collect information about logarithmic designs and give us a mathematical methodology to calculate the hidden properties of different certain computer networks and other constructs. Different techniques are accessible history to look at the idea of a topological index. There are two chief contentions of topological indices, first one is the degree based topological and the sub-optimal is known as distance-based topological indices. There are numerous such invariants are accessible in history [24]. Irregularity Sombor Index has an extraordinary capacity of assumption in the field of computer science, math, chemistry, drugs, informatics and power age in setting with physical and substance designs and organizations.

ISO index stands for Irregularity Sombor Index which is introduced by V. R Kulli after taking inspiration from Sombor Indices. ISO index has the quality to predict the hidden properties of a network and find the lower bounds, upper bounds and irregularities from the existing networks. The deduced results would be used for the modeling of certain computer networks, their up gradation with best characteristics, finding new network architectures and also used as guidelines for the developments of advanced networks used in different fields of computer science and other sciences

Comment 4: (3) The authors also should do a comparison with the previous reports to the advantage of this manuscript.

Response 4: We compare different studies with our present study in the following table. After analysis it is found that as compared to other we found irregularities from the existing networks which was the major issue and hurdle in efficiency, performance and security of the networks. It also found sharp upper bounds and lower bound

Sr. No.

Title of Research Paper

Year

Networks Solved

Invariants Used

Results

1

Topological Properties Of Degree-Based Invariants Via M-Polynomial Approach

2022

Hexagonal Networks

First And Second Zagreb Indices, Modified First Zagreb Index, Nano-Zagreb Index, Second Hyper-Zagreb Index, Randi´C

Index,Reciprocal Randi´C Index, First Gourava Index, and Product Connectivity Gourava Index

Analyze and gave valuable information about the molecular

structure or network and applications in QSPR & QSAR.

2

Contraharmonic Quadratic Index Of Certain Nanostar Dendrimers

2022

Dendrimer Nanostars

Contraharmonic-Quadratic Index and Quadratic-Contraharmonic Index

computed the CQ index for some standard graphs

3

Some Results On The Sombor Indices of Graphs

2021

Degree-Regular Graph/Network

The Sombor Index, The

Reduced Sombor Index and the Average Sombor Index

Establishing inequalities related to the

aforementioned three graph invariants and proving a recently proposed conjecture concerning the sombor index

4

Some Basic Properties of Sombor Indices

2021

Regular Graph or Network

Vertex–Degree–Based (VDB) Molecular

Structure Descriptors (Sombor Index and its Reduced Form)

Any reduced VDB index can be viewed as a reduced

sombor-type index

5

Analysis Of Dendrimer Generation By Sombor Indices

2021

Dendrimers Generation Networks

Sombor Index and Reduced Sombor Index

Computed the newly introduced sombor indices for phosphorus-

containing dendrimers, porphyrin-cored dendrimers,

pdi-cored dendrimers, triazine-based dendrimers,

and aliphatic polyamide dendrimers.

6

Sombor Index of Some Nanostructures

2021

Nanostructures

Sombor Index

Computed explicit formulae for sombor index

of 2D-lattice, nanotube, and nanotorus

7

Polynomials And General Degree-Based Topological Indices of

Generalized Sierpinski Networks

2021

Sierpinski Networks

Connectivity Polynomials Such Asm-Polynomial, Zagreb Polynomials, Forgotten Polynomial, (Α, Β)-Zagreb Index and Several Other General Indices

These facts can be used as raw inputs in

networking and fractals where these networks play a

significant role. physicochemical

properties of the molecules modeled on the

S(k, n) networks can be forecasted using the results

.

8

The Calculations of Topological Indices on Certain Networks

2021

Hexagonal Networks

ABC Index, AZI Index, GA Index, The

Multiplicative Version Of Ordinary First Zagreb Index, The Second Multiplicative Zagreb Index, and Zagreb Index

By

calculating the correlation index of several specific chemical

networks, a study can get the above indices formulas. 0is also

provides potential help for scholars to study networks

characteristics better. for further work, if the corresponding

networks are replaced by other networks

9

Discovering Irregularities from Computer Networks by Topological Mapping

2022

Bridge Networs, Hexagonal Networks, Honeycomb Networks and Sierpinski Networks,

Irregularity Sombor Index

Finding Sharp upper bounds, lower bounds and irregularities

Comment 5: (4) Fig.5 is not clear, and the authors can consider to replote it.

Response 5: Its topology is clear now 

Comment 6: (5) The authors can do a short introduction about the development in data acquisition or sensing. Especially, the topology technology is used in the sensing. Some references can be considered to add in this manuscript. For instance,

#B.F. Wan et al. in IEEE Sensors Journal, vol. 21, no. 1, pp. 331-338, 1 Jan.1, 2021, doi: 10.1109/JSEN.2020.3013289.

Response 6: This is added and cited

 Hamid, K.; Iqbal, M. waseem; Muhammad, H.; Fuzail, Z.; Nazir, Z. ANOVA Based Usability Evaluation Of Kid’s Mobile Apps Empowered Learning Process. Qingdao Daxue Xuebao(Gongcheng Jishuban)/Journal of Qingdao University (Engineering and Technology Edition) 2022, 41, 142–169

Comment 7: Good luck with the corrections and I am looking forward to reading the updated version of your article. 

Response 7: Thank you very it is highly encouraging and fruitful for us

Round 2

Reviewer 3 Report

The authors satisfactorily addressed the issues raised by the reviewers. So I recommend that the revised manuscript can be accepted for publication